# Molecular Signaling to Preserve Mitochondrial Integrity against Ischemic Stress in the Heart: Rescue or Remove Mitochondria in Danger

**DOI:** 10.3390/cells10123330

**Published:** 2021-11-27

**Authors:** Justin D. Yu, Shigeki Miyamoto

**Affiliations:** Department of Pharmacology, University of California San Diego, 9500 Gilman Drive, La Jolla, CA 92093, USA; jdy002@ucsd.edu

**Keywords:** mitochondria, cell death, BCL-2 family proteins, mPTP, pro-survival signaling, mitophagy, ischemia, heart

## Abstract

Cardiovascular diseases are one of the leading causes of death and global health problems worldwide, and ischemic heart disease is the most common cause of heart failure (HF). The heart is a high-energy demanding organ, and myocardial energy reserves are limited. Mitochondria are the powerhouses of the cell, but under stress conditions, they become damaged, release necrotic and apoptotic factors, and contribute to cell death. Loss of cardiomyocytes plays a significant role in ischemic heart disease. In response to stress, protective signaling pathways are activated to limit mitochondrial deterioration and protect the heart. To prevent mitochondrial death pathways, damaged mitochondria are removed by mitochondrial autophagy (mitophagy). Mitochondrial quality control mediated by mitophagy is functionally linked to mitochondrial dynamics. This review provides a current understanding of the signaling mechanisms by which the integrity of mitochondria is preserved in the heart against ischemic stress.

## 1. Introduction

Heart failure is a clinical syndrome characterized by an inadequate blood supply resulting from impaired cardiac pump function and is often the end-stage manifestation of cardiovascular diseases [1]. Ischemia, a condition in which the blood flow is restricted, is the principal etiology of heart failure. During ischemia, the supply of oxygen and nutrients to cardiomyocytes is limited, leading to energy depletion and eventual cardiomyocyte death. Although re-establishing blood flow (reperfusion) is essential for salvaging viable myocardia, reperfusion paradoxically elicits cell death (I/R injury) [2]. Cardiomyocytes are highly differentiated cells with limited regenerative capacity, thus irreversible cell death plays a crucial role in ischemic heart disease [3,4,5,6].

As a result of the symbiotic invasion of eukaryotes by eubacteria more than 1.5 billion years ago, mitochondria have evolved to become the critical organelles for the generation of cellular energy [7]. Cardiac muscle is especially abundant in mitochondria to meet the high energy demands of beat-to-beat contraction. Mitochondria are double-membrane organelles consisting of inner and outer mitochondrial membranes, occupying 30% of the total cardiomyocyte volume, and generating up to 30 kg of ATP per day [8]. It is on the mitochondrial inner membrane (MIM) that oxidative phosphorylation (OXPHOS) takes place to generate ATP, and this process is driven by the mitochondrial membrane potential across the MIM. Mitochondria also play a pivotal role in cell death in response to cellular stress [4,5,6]. BCL-2 family proteins and the mitochondrial permeability transition pore (mPTP) are important regulators of the mitochondrial death pathway. Apoptotic BCL-2 family proteins such as BAX, BAD, and BID induce mitochondrial outer membrane (MOM) permeabilization (MOMP), leading to the release of apoptotic molecules such as cytochrome c from the intramembrane space (IMS) which in turn activates caspase-9/caspase-3 and induces apoptosis. Although the molecular composition of the mPTP has not been entirely clear, reactive oxygen species (ROS) and mitochondrial calcium overload induce the opening of the mPTP at the inner membrane, causing mitochondrial depolarization, rupture of the outer mitochondrial membranes, and necrotic cell death. As many stress signal pathways converge on mitochondria to induce cell death, preservation of mitochondrial integrity is essential to the survival of cardiomyocytes against stress. 

Protective signaling molecules are recruited in response to stress to prevent the mitochondrial death pathways. The activation of protective protein kinases such as Akt and PKCε leads to the inhibition of apoptotic BCL-2 family proteins and/or those of mPTP. Inhibition of upstream stress signaling, for example, the attenuation of mitochondrial ROS generation, is also protective. In addition to inhibition of the mitochondrial death pathways, mitochondrial integrity is also preserved by mitochondrial quality control mechanisms. Mitochondria-specific autophagy (mitophagy) removes damaged mitochondria through lysosomal degradation, preventing the accumulation of toxic mitochondria (Figure 1).

## 2. Mitochondrial Death Pathway

### 2.1. BCL-2 Family Protein and Apoptosis

The BCL-2 family proteins are divided into three groups based on their function. (1) Pro-apoptotic proteins, which form a pore on MOM (BAX, BAK). These pro-apoptotic proteins contain three BCL-2 homology domains (BH-domain; BH1, BH2, and BH3). (2) Pro-apoptotic BH3-only proteins (BAD, BID, BIK, BIM, BMF, HRK, NOXA, PUMA, BNIP3, NIX/BNIP3L), and (3) pro-survival proteins which contain BH1-4 domains. The BH4 domain is critical to gain the protective effect (BCL-2, BCL-XL, BCL-W, MCL-1, BFL-1) [9,10,11] (Figure 2A). 

BAX and BAK are activated in response to stress, translocating to the mitochondria and forming an oligomeric pore, thus permeabilizing the mitochondrial outer membrane. Thus, BAX and BAK are executors of MOMP [11,12,13]. MOMP leads to the release of cytochrome c and the subsequent formation of the apoptosome, a protein complex that activates caspase-9. The activated caspase-9 then cleaves and activates the effector caspases, including caspase-3 [14]. The BAX/BAK-mediated MOMP is inhibited by pro-survival BCL-2 family proteins and positively regulated by BH3-only proteins [9,10,15,16]. Pro-survival BCL-2 family proteins and BH3-only proteins interact with each other, resulting in mutual sequestration and inhibition of their pro-survival and apoptotic effects. Pro-apoptotic BH3-only proteins are classified as activator or sensitizer proteins based on whether they directly bind to and activate BAX and BAK [11]. BID, BIM, and PUMA are the activators, whereas other BH3-only proteins are sensitizers. Activator BH3 only proteins bind to BAX and BAK, triggering oligomerization of BAX/BAK and MOMP. Activator BH-3 only proteins also bind and inhibit pro-survival BCL-2 family proteins. Sensitizer BH3-only proteins bind to pro-survival proteins to prevent their inhibitory binding to BAX and BAK (Figure 2A). 

The significance of BCL-2 family proteins in ischemic injury in the heart has been demonstrated [17,18,19,20,21,22]. Specifically, BCL-2 overexpression provides cardioprotection against ischemia/reperfusion (I/R) in vivo, and BAX KO mice show smaller I/R damage in the perfused heart [17,19]. I/R-induced Ca^2+^ increases and subsequent activation of calpain can also lead to cleavage of BID, producing t-BID to induce apoptotic cell death [18]. The expression of BNIP3 is increased by hypoxia and elicits apoptotic cell death induced by simulated I/R in cardiomyocytes [20]. The expression of NIX (BNIP3L) is increased by the stimulation of Gq, a large G-protein, and triggers apoptosis in the heart [22]. Global deletion of PUMA, a BH3-only protein transcriptionally regulated by p53, decreases the infarct size and improves cardiac function in the perfused mouse heart subjected to I/R [21]. 

### 2.2. Mitochondrial Permeability Transition Pore (mPTP)

mPTP is a high-conductance and non-specific channel located in the mitochondrial inner membrane. Cardiac injury induced by I/R results from increased cytosolic Ca^2+^ and the generation of reactive oxygen species (ROS) [4,6,23,24] (Figure 2B). As a result of loss of ionic homeostasis during ischemia, supra-physiological levels of cytoplasmic Ca^2+^ lead to mitochondrial Ca^2+^ overload during reperfusion, leading to the opening of the mPTP. Ca^2+^ induced the mPTP opening is potentiated by ROS, loss of adenine nucleotides, and increases in phosphate. Opening of the mPTP causes mitochondrial swelling, rupture of mitochondrial membranes, and necrotic cell death. 

The molecular constituents of the mPTP continue to be disputed. Historically, mPTP was generally depicted as a core protein complex composed of a voltage-dependent anion channel (VDAC), an adenine nucleotide transporter (ANT), and a mitochondrial phosphate carrier (PiC); however, studies using gene knockout or knockdown revealed that these molecules are not essential components of the mPTP [4,25,26,27]. Although not fully established, recent evidence suggests that the dimer of F_0_-F_1_ ATP synthase could be the pore-forming core component of the mPTP [28,29]. Most recently, SPG7 has been reported as a main component of the mPTP [30], but it could be a regulator instead [31].

Irrespective of the molecular composition of the mPTP, cyclophilin D (Cyp-D), an inner mitochondrial membrane protein, is a critical positive regulator of the mPTP. A regulatory role of Cyp-D to mPTP-opening was first suggested by the inhibitory effect of cyclosporin A (Cs-A) on Ca^2+^-induced mPTP opening and by the subsequent discovery that Cs-A binds to Cyp-D [32,33]. It has recently been reported that the mPTP sensitizing effect of Cyp-D is potentiated by the phosphorylation of Cyp-D at Ser191 [34]. It would be of importance to identify responsible kinases for this phosphorylation (Figure 2B). It has been demonstrated that Cyp-D KO mouse hearts exposed to I/R show significantly less injury than WT mice, whereas Cyp-D-overexpressing mice show mitochondrial swelling and spontaneous cell death, suggesting that Cyp-D sensitizes mitochondria to Ca^2+^ induced mPTP opening [3,35]. 

Mitochondrial hexokinase II (mitoHKII), a glycolytic kinase, has been suggested to be a negative regulator of the mPTP. It has been shown that an increase in mitoHKII has an inhibitory effect on Ca^2+^- and ROS-induced mPTP opening and that a large dissociation of mitoHK-II sensitizes the mPTP opening and ischemic cardiac injury. The mechanisms by which HKII inhibits the mPTP could be involved in the direct inhibitory effect on the mPTP or indirect inhibition through regulation of mitochondrial ROS [36,37,38,39,40]. Heterozygote HKII KO mice have normal cardiac function at baseline; however, they have a higher degree of damage after I/R compared to WT [41]. Furthermore, mitochondrial HKII dissociating peptide treatment significantly decreases mitochondrial HKII levels without affecting baseline cardiac function, but dramatically increases ischemia-reperfusion injury [40].

### 2.3. Interaction between Cell Death Pathways

Although it has been disputed whether BAX/BAK regulates the mPTP opening, genetic studies revealed that deletion of BAX/BAK inhibits both the mPTP opening and necrotic cell death, suggesting the ability of BAX/BAK to facilitate mPTP-induced necrotic cell death [42,43]. The NLRP3 inflammasome is a multiprotein complex consisting of NLRP3, the adaptor protein ASC and pro-caspase-1 and it plays a critical role in sensing cellular stress and eliciting inflammation by the caspase-1 dependent maturation of IL-1β and IL-18 [44,45]. Recent studies have shown that the NLRP3 inflammasome plays a critical role in cardiac pathology, including ischemic heart disease [45,46,47,48,49,50]. Caspase-1 also induces cell death (pyroptosis) by the caspase-1-mediated cleavage of gasdermin D (GSDMD) and the resultant pore formation in the plasma membrane [51,52]. It has been demonstrated that caspase-1 inhibition provides protection against MI [53]. In addition to pyroptosis, a recent study in non-cardiac cells revealed that caspase-1 activation can induce apoptosis by the cleavage and activation of BID [54], suggesting an interaction between the inflammasomes and mitochondrial apoptosis. 

## 3. Signaling Pathways to Regulate the Mitochondrial Death Pathways

### 3.1. Protective Signaling to Inhibit the Mitochondrial Death Pathways

#### 3.1.1. PKCε

There are many intracellular signals involved in the prevention of the mitochondrial death pathways. Protein kinase C (PKC) was one of the first identified to prevent mitochondrial deterioration during ischemic stress. There are many PKC isoforms expressed in cardiomyocytes, and PKCε has been demonstrated to be cardioprotective [55,56,57]. Earlier studies suggested that activated PKCε translocates to the mitochondria, interacting with and inhibiting the mPTP opening (Figure 2B) [55]. It has also been suggested that PKCε is imported into the mitochondria in an HSP90 dependent manner [58] and that PKCε phosphorylates and activates mitochondrial aldehyde dehydrogenase 2 (ALDH2) in the mitochondrial matrix, leading to the inhibition of ROS generation [59,60]. PKCε-mediated ROS inhibition in the mitochondria could also be explained by activation of the mitochondrial K_ATP_ channel [61,62]. However, it should be pointed out that there is no universal agreement in the literature on the presence of mitoKATP channels [63]. 

PKCε has also been implicated in the regulation of apoptosis. PKCε phosphorylates BAD at Ser112 to inhibit MOMP and apoptosis [64], although this is not established in the heart (Figure 2A). 

The administration of a selective activator peptide of PKCε reduces infarct size and results in fewer cases of ventricular fibrillation during I/R in pigs [65]. PKCε also contributes to cardioprotection mediated by ischemic preconditioning, a series of brief cycles of I/R before a sustained period of ischemia [56].

#### 3.1.2. Akt

Akt is activated downstream of phosphatidylinositol 3-kinase (PI3K) in response to stimulation of receptor tyrosine kinases, G-protein coupled receptors (GPCRs), and oxidative stress. Akt exerts a strong cardioprotective effect against ischemic stress [36,66,67,68,69]. Akt activated at the plasma membrane is active in the cytosol, and it also translocates to subcellular compartments including the nucleus and mitochondria [36,70,71]. Earlier studies demonstrated that Akt inhibits apoptosis through phosphorylation of BAD at Ser136 [72,73]. Another BCL-2 family protein regulated by Akt is BAX, which is a key molecule in MOMP. Phosphorylation of BAX by Akt at Ser184 leads to inhibition of the conformational change required for BAX translocation to the mitochondria [74,75] (Figure 2A). 

The opening of the mPTP is also negatively regulated by Akt. Akt phosphorylates HKII at Ser473 and increases mitochondrial HKII association, leading to inhibition of the mPTP opening [36,68,76,77] (Figure 2B). Intracellular Ca^2+^ overloading, the main trigger of mPTP opening, is also normalized by Akt activation through transcriptional regulation of Na^+^/Ca^2+^ exchanger [78]. The expression of constitutively active Akt protects the heart against I/R [66], and the inhibition of PH domain leucine-rich repeat protein phosphatase-1 (PHLPP1), an Akt phosphatase, increases Akt activity and reduces infarct size in a mouse Langendorff model of I/R [79]. Many studies have also demonstrated that activation of Akt contributes to the cardioprotective effects of receptor ligands, including insulin-like growth factor-1 (IGF-1), leukemia inhibitory factor (LIF), and sphingosine-1-phosphate (S1P) [66,68,79,80]. Akt is a converging point of many protective signaling pathways, as mentioned below. PIM1 kinase has been suggested to function as a downstream effector of Akt [70,81]. 

#### 3.1.3. RhoA

RhoA is a small G-protein that transduces extracellular signals into a range of cellular responses such as cell survival and proliferation [82,83,84]. RhoA is activated in response to stimulation of various GPCRs including those for S1P and lysophosphatidic acid (LPA) [84,85,86]. RhoA is also directly activated by ROS [87]. RhoA signaling stimulates various survival pathways including Akt, focal adhesion kinase (FAK), and protein kinase D (PKD) [86,88,89,90]. FAK activation by RhoA leads to Akt activation, contributing to RhoA mediated protection [81]. PKD is activated through the RhoA-mediated activation of phospholipase Cε (PLCε), leading to PKD-dependent phosphorylation and inhibition of the cofilin phosphatase slingshot 1L (SSH1L). Inhibition of SSHL1 results in an increase in phosphorylated cofilin, thereby inhibiting cofilin-dependent BAX translocation to the mitochondria (Figure 2A) [89]. PKN is another downstream kinase of RhoA that provides protection against I/R [91]. RhoA also regulates gene transcription through YAP and MRTF-A, transcriptional co-activators [92,93,94], both of which have been reported to be cardioprotective. MRTF-A regulates the expression of CCN1, which binds to integrin receptors and activates Akt to provide protection against I/R [86], while YAP activation leads to upregulation of Akt signaling and provides cardioprotection against MI [95]. Although an earlier study showed that overexpression of active RhoA in the heart is deleterious [96], physiological levels of RhoA activation have been demonstrated to be cardioprotective against ischemic and non-ischemic stresses [88,89,90,91,97]. It has also been shown that cardiac-specific RhoA KO enhances I/R injury in the heart [90]. 

#### 3.1.4. PKG

Many preclinical studies have suggested a cardioprotective role of nitric oxide (NO) signaling, revealing cGMP-dependent protein kinase (protein kinase G:PKG) as the responsible signaling pathway [98,99]. NO activates soluble guanylate cyclase (sGC) and ANP/BNP activates membrane-bound GC (pGC), leading to GC-dependent formation of cGMP [98,100]. PKG is reported to block mPTP in cardiac mitochondria [101]. Mechanistically, PKG has been linked to the activation of the mitochondrial K_ATP_ channel [102] and a recent paper further suggests that PKG activates the cardiomyocyte-specific BK channel at the mitochondria, increasing K^+^ influx into the matrix, opposing mitochondrial Ca^2+^ overloading and ROS production [103]. PKG can diminish Ca^2+^ overloading through the inhibition of the Na^+^/H^+^ exchanger on the plasma membrane [104,105]. The protective effects of the NO/cGMP/PKG pathway against ischemic stress in the heart have been repeatedly demonstrated. Activation of sGC provides cardioprotection against I/R in vivo and in vitro [106,107,108]. Pharmacological activation of PKG reduces infarct size and preserves cardiac contractility in rat hearts subjected to I/R [106]. The protective effect of PKG is further supported by a recent study demonstrating that suppression of PKG activity by cardiac-specific expression of PDE5A worsens infarct and cardiac remodeling in MI hearts [107]. 

#### 3.1.5. STAT3

Signal transducer and activator 3 (STAT3) is a member of the STAT family of nuclear transcription factors and is activated through phosphorylation by Janus kinases (JAKs) [109,110,111]. The JAK/STAT3 pathway is regulated by diverse receptors including gp130 and GPCRs. It has been demonstrated that STAT3 is protective in the heart against stress, including I/R, and that STAT3 is activated by preconditioning, contributing to its cardioprotective effect [112,113]. Although it is still somewhat controversial [111], STAT3 localizes to the mitochondria in the heart [109]. Mitochondrial STAT3 inhibits the mPTP opening, and this effect is suggested to be mediated by STAT3 binding to Cyp-D (Figure 2B) [114,115,116]. Mitochondrial STAT3 also protects complex I against ischemic damage and thereby decreases ROS production from complex I [116]. STAT3 deficient mice show enhanced susceptibility to I/R injury with increased cardiac apoptosis and reduced cardiac function [117]. Intriguingly, cardiac-specific STAT3 ablation in the sub-acute phase aggravated the survival rate after MI [118]. 

#### 3.1.6. SIRT3 

Sirtuins (SIRT1-7) are NAD^+^-dependent deacetylases that regulate a wide variety of cellular processes, including metabolism, mitochondrial homeostasis, and oxidative stress. SIRT3, SIRT4, and SIRT5 are primarily localized at the mitochondria and control the acetylation status of mitochondrial proteins, regulating mitochondrial function [119,120,121]. Emerging evidence has suggested a SIRT3 role in a variety of cardioprotective mechanisms. Mechanistically, SIRT3 is shown to interact with and deacetylate Cyp-D at Lys166, inhibiting the mPTP opening [122,123] (Figure 2B). SIRT3 can also negatively regulate the opening of the mPTP by increasing antioxidant capacity through the upregulation and activation of manganese superoxide dismutase (MnSOD), a major mitochondrial antioxidant enzyme [124,125,126]. SIRT3 exerts anti-apoptotic effects which are mediated by the deacetylation of Ku70 to inhibit BAX translocation to the mitochondria [127]. Studies using SIRT3 heterozygous or homozygous KO mice demonstrated that decreased SIRT3 increases the susceptibility of the heart to I/R injury [123,128], although there are conflicting data [129]. The difference observed in the effect of SIRT3 on I/R injury could be explained by age differences: the significance of SIRT3 in protection is increased with aging [121].

### 3.2. Stress Signaling to Enhance the Mitochondrial Death Pathways

#### 3.2.1. GSK-3β

Glycogen synthase kinase 3 (GSK-3β) was initially discovered in the context of glycogen synthesis [130]. GSK-3β is active in basal conditions, and upstream kinases including Akt phosphorylate GSK-3β at Ser9 and inhibit its activity [131,132]. An early in vitro study revealed that GSK-3β exerts a stimulatory effect on mPTP opening [133] (Figure 2B). Thus, the protective effects of Akt on the preservation of mitochondrial integrity could be explained by the inhibition of GSK3β mediated by Akt. 

GSK3β has been demonstrated to phosphorylate VDAC1 at Thr51, which results in the disruption of HKII binding to the mitochondria, potentiating cell death [134]. GSK3β enhances apoptotic cell death through the regulation of BCL-2 family proteins. GSK3β phosphorylates BAX at Ser163 facilitating BAX translocation to the mitochondria [135] and also phosphorylates MCL-1 at Ser159, inducing MCL-1 degradation [136]. 

Pharmacological and genetic studies have demonstrated that inhibition of GSK-3β activity is protective against I/R, suggesting that GSK-3β contributes to the development of cardiac injury induced by I/R [137,138,139,140]. On the contrary, GSK3β is shown to be protective against MI in the heart, and this is suggested to be mediated by the stimulation of general autophagy induced by GSK3β [141]. 

#### 3.2.2. Mst1

Mst1 (mammalian Ste20-like kinase 1) is a serine-threonine kinase and a component of the Hippo signaling pathway, which regulates cell survival and organ size [93,94,142]. Mst1 translocates to the mitochondria in response to oxidative stress and phosphorylates BCL-xL at Ser14, antagonizing BCL-xL-BAX binding. This leads to BAX activation and subsequent apoptosis [143]. Mst1 also negatively regulates YAP, a transcriptional cofactor, through Lats1/2 (large tumor suppressor 1/2) activation. YAP facilitates forkhead box O1 (FoxO1) dependent expression of MnSOD, an antioxidant molecule. Thus, Mst1 activation increases oxidative stress through the inhibition of YAP, which could enhance the opening of the mPTP and contribute to I/R injury [144]. Mst1 is highly active during apoptosis, and inhibition of Mst1 protects the heart against I/R and MI [145,146]. The significance of BCL-xL phosphorylation at Ser14 by Mst1 in I/R injury is further supported by studies using the adenoviral expression of non-phosphorylatable mutant BCL-xL (BCL-xL S14A) in the heart and BCL-xL S14A knock-in mice [145,146] (Figure 2A). 

## 4. Mitochondrial Quality Control by Mitophagy

### 4.1. Conventional PINK1/Parkin Mediated Mitophagy

Autophagy is an evolutionarily conserved catabolic process that is rapidly induced in response to stress. Cytoplasmic components and damaged organelles are engulfed by autophagosomes, followed by fusion with lysosomes and degradation by lysosomal enzymes [147,148,149,150]. Preservation of mitochondrial quality is fundamentally important in maintaining cellular homeostasis, as damaged mitochondria induce apoptotic and necrotic cell death. Thus, there exists a process for the selective elimination of damaged mitochondria by autophagy (mitophagy) [151,152,153,154]. Although excessive mitophagy can induce cell death, including autosis, [155,156,157], it is generally accepted that a moderate level of mitophagy plays a protective role against stress in the heart. 

One of the best-established mechanisms of mitophagy is the mitochondrial membrane depolarization-dependent PINK1/Parkin pathway (Figure 3). It has been demonstrated that the PINK1/Parkin-mediated mitophagy is protective in the heart. For instance, PINK1 KO mice display larger infarct size in response to I/R [158], and Parkin KO mice present more damage after MI [159]. 

In healthy cells, PINK1, a mitochondrial serine/threonine kinase, is partially imported into the mitochondria and embedded into the mitochondrial inner membrane. The transmembrane segment of PINK1 is cleaved by an inner-mitochondrial membrane protease, and the cleaved PINK1 is then retro-translocated to the cytosol, where it undergoes proteasomal degradation [151,160,161,162,163] (Figure 3A). Under stress conditions, when survival signaling fails to prevent mitochondrial damage, dissipation of the mitochondrial membrane potential leads to inhibition of PINK1 cleavage and supports the accumulation of full-length PINK1 at the MOM [162,163,164,165] (Figure 3B). Full-length PINK1 forms a high-weight molecular complex with the TOM complex on the MOM. This molecular complex contains two PINK1, leading to trans-phosphorylation-dependent activation of PINK1 [163,166,167]. Activated PINK1 phosphorylates basal MOM ubiquitin which drives Parkin recruitment to the mitochondria [168]. PINK1 further phosphorylates Parkin to stabilize the binding of Parkin to ubiquitin, Parkin ubiquitinates outer mitochondrial membrane proteins, and ubiquitin is further phosphorylated by PINK1, generating positive feedback effects [162,163,169,170]. In addition, PINK1 phosphorylates mitofusin-2 (MFN2), a mitochondria fusion protein that also serves as a receptor for Parkin at MOM [153,171]. Ubiquitinated mitochondria are recognized by the autophagosome membrane through the autophagy receptor proteins, which interact with LC3 (ATG8) on the autophagosome membrane. The first autophagy receptor discovered in mammalian cells was p62, but further studies revealed that p62 is not essential for mitophagy. Other autophagy receptors are the next to BRCA1 gene 1 protein (NBR1), calcium-binding and coiled-coil domain-containing protein 2 (NDP52), optineurin (OPTN), and Tax1-binding protein 1 (TAX1BP1) [152,153,154]. A study using a cell line lacking all these receptors suggested that NDP52 and OPTN are the primary, yet redundant, receptors [172].

Although regulation of mitophagosome formation has been elusive, recent evidence suggests that this is regulated by Rab proteins, key regulators of membrane trafficking. Parkin-mediated ubiquitinated mitochondria recruit RABGEF1, an activator of Rab proteins, that regulates Rab5 translocation to the mitochondria. Rab5 then directs Rab7 to the mitochondria, and Rab7 facilitates LC3-labeled pre-autophagosome expansion. Termination of Rab activity by the TBC1D15/17 RabGAPs is required for the autophagic encapsulation of mitochondria [173,174,175,176]. A recent study demonstrated that TBC1D15 expression in the heart provides protection against MI through the enhancement of mitophagy flux [177]. 

#### 4.1.1. Positive Regulators of the PINK1/Parkin Pathway 

##### BAG2, BAG3, and BAG6

Accumulating evidence suggests that there are signaling molecules that regulate PINK1/Parkin mediated mitophagy (Figure 3C). BCL-2-associated athanogene (BAG) family protein consists of 6 members (BAG1–6), all sharing the BAG protein domain, through which they bind to the heat shock protein 70 (Hsp70), acting as a co-chaperone. BAG family proteins regulate protein quality through proteasomal and autophagic protein degradation [178,179,180]. BAG3 is highly expressed in the skeletal and cardiac muscles, and mutations in the BAG3 gene cause cardiomyopathy [181,182,183]. BAG3 gene deletion in mice leads to lethal myopathy, and overexpression of BAG3 protects the heart against I/R through the regulation of autophagy [184,185]. BAG3 has been shown to translocate to the mitochondria in response to mitochondrial membrane depolarization and positively regulate mitophagy in cardiomyocytes, possibly through the regulation of Parkin [186].

BAG2 is also reported to promote mitophagy by inhibiting PINK1 degradation in neurons [187,188], although the role of BAG2 in the heart has not been determined. A recent study demonstrated that BAG6 localizes in the mitochondria matrix but translocates to the MOM after mitochondrial depolarization to enhance PINK1 accumulation at the mitochondria [189]. In addition, the authors showed that BAG6 functions as a mitophagy receptor by binding to LC3, facilitating mitophagy in cancer cells. It would be of interest to determine whether BAG6 also regulates mitophagy in the heart. 

##### SIRT3

The acetylation/deacetylation of mitochondrial proteins has a significant impact on mitochondrial function, including mitophagy. It has been established that SIRT3, a mitochondrial SIRT, functions as a protective molecule in the heart against stress, including MI, as discussed above, and the activation of mitophagy is involved in SIRT3 mediated cardioprotection. An initial study in HEK293T cells demonstrated that SIRT3 binds to ATP synthase but dissociates from it upon mitochondrial membrane depolarization, leading to deacetylation of mitochondrial proteins [190]. Recent studies have shown that SIRT3 directly deacetylates PINK1 and Parkin to induce mitophagy [191,192]. As an additional mechanism, SIRT3 deacetylates and activates FoxO3A, leading to transcriptional upregulation of Parkin and the activation of mitophagy to protect the heart [193]. As mentioned earlier, previous studies using SIRT3 hetero-and homozygous KO mice demonstrate the protective role of SIRT3 in the heart against ischemic stress. The contribution of mitophagy to SIRT3-induced cardioprotection has not, however, been determined. 

##### PTENα (PTEN-L)

Phosphatase and tensin homolog (PTEN) is a phosphatase that is a well-known tumor suppressor which dephosphorylates proteins and lipids [194,195]. The lipid phosphatase activity, converting PIP_3_ to PIP_2_, is the most characterized function, suppressing Akt activity. PTENα is a new isoform that contains an alternatively translated region at its N-terminus and is localized predominantly in the cytoplasm and mitochondria [196,197]. Mitochondrial PTENα plays a significant role in preserving mitochondrial function and energy metabolism [197]. Mechanistically, PTENα promotes Parkin translocation to the mitochondria through its binding to Parkin upon mitochondrial depolarization. Interestingly, a study in HeLa cells reached the opposite conclusion, that PTENα negatively regulates mitophagy through dephosphorylation of Ub [198]. The exact reason for the discrepancy has not been clear. Nonetheless, PTENα-deficient mice develop cardiac mitochondrial dysfunction. In adult ventricular myocytes isolated from PTENα KO mice, CCCP-induced mitophagy is impaired and PTENα is suggested to positively regulate Parkin mitochondria translocation by enhancing Parkin self-association in a phosphatase-independent manner [199]. 

##### TBK1

The ubiquitinated mitochondria recruit autophagy/mitophagy receptors such as OPTN and NDP52, which interact with LC3 on the autophagosome membrane. TANK-binding kinase 1 (TBK1) is recruited to the mitochondria and phosphorylates OPTN at S473 to stabilize OPTN binding to Ub chains [200]. Moreover, the TBK1 mediated phosphorylation of OPTN at S177 promotes the binding of OPTN to LC3II on the autophagosome, facilitating the recognition of damaged mitochondria by autophagic machinery [201]. Similarly, TBK1 is shown to be required for efficient retention of NDP52 and p62 on damaged mitochondria [202,203]. Although the role of TBK1 in cardioprotection against ischemic stress has not been well studied, recent studies have demonstrated that TBK1 provides cardioprotection against I/R and MI, in which inhibition of apoptosis and increased mitophagy flux through the TBK1-NDP52 pathway play a critical role [204,205]. 

##### AMPK1/ULK1

AMP-activated protein kinase (AMPK) is activated by the reduction in cellular ATP levels (increase in AMP/ATP ratio) and positively regulates autophagy through the inhibition of mTOR complex1 and the activation of unc51-like kinase 1 (ULK1) [206,207]. The ULK1 complex, composed of ULK1 (Atg1), Atg13, Atg101, and FIP200, plays a critical role in vesicle nucleation in autophagy. In addition to these well-established roles, recent studies indicate that AMPK and ULK1 directly regulate mitophagy [207]. 

Myocardial injury and apoptotic activity following low-flow I/R are increased in mice expressing a kinase-dead form of AMPK, suggesting the protective role of AMPK against energetic stress in the heart [208]. A direct link between AMPK and mitophagy has been demonstrated. A study using AMPKα2 KO mice suggested that AMPKα2 phosphorylates PINK1 at Ser495 to enhance mitophagy and prevent the progression of heart failure induced by pressure overload [209]. Interestingly, mitochondria-localized AMPK has recently been discovered, and it is suggested that a mitochondrial pool of AMPK responds to local energetics stresses, including cardiac ischemia, to induce mitophagy [210]. 

A recent in vitro study revealed that ULK1 phosphorylates Parkin at Ser108 in response to mitochondrial stress, positively regulating Parkin translocation to the mitochondria and mitophagy [211]. ULK1 has also been suggested to initiate mitophagy independent of LC3. Specifically, TBK1/NDP52 on ubiquitinated mitochondria recruits the ULK1 complex to the mitochondria and the ULK1 complex initiates autophagosome biogenesis directly on mitochondria [212]. ULK1 has been demonstrated to regulate alternative autophagy in the heart [213], as discussed below, and it would be of interest to determine whether the multiple roles of ULK1 in the regulation of mitophagy play distinct roles in the heart in response to different stresses. 

#### 4.1.2. Negative Regulators of the PINK1/Parkin Pathway

##### Pro-Survival Members of BCL-2 Family Proteins

It has been reported that BCL-2 family proteins interact with and negatively regulate Parkin or PINK1. A study in HeLa cells demonstrated that Parkin translocation induced by mitochondrial depolarization is antagonized by pro-survival BCL-2 family proteins, including BCL-xL and MCL-1, by direct interaction, and this interaction is negatively regulated by BH3-only proteins including BAD, BIM, NOXA, and PUMA [214]. This inhibitory effect on Parkin of pro-survival BCL-2 proteins was further confirmed by recent studies [215,216]. Specifically, BCL-xL inhibits mitophagy by binding to Parkin in the cytosol resulting in the inhibition of Parkin recruitment to the mitochondria [216]. Thus, pro-survival BCL-2 family proteins operate to inhibit apoptosis as well as to keep mitophagy in check. Interestingly, PINK1 and Parkin also regulate BCL-2 family proteins. PINK1 phosphorylates BCL-xL to prevent its cleavage at depolarized mitochondria, and Parkin inhibits BAK and BAX apoptotic function during mitophagy to suppress errant apoptosis [217,218]. It appears that pro-survival BCL-2 family proteins prevent the PINK1/Parkin pathway, while the PINK1/Parkin pathway positively regulates the pro-survival effects of BCL2 family protein. Thus, pro-survival BCL-2 family proteins block the unnecessary removal of mitochondria, but when PINK1 and Parkin are activated, they support mitochondrial protection through the regulation of BCL-2 family proteins. The functional interplay between BCL-2 family proteins and the PINK1/Parkin pathway remains to be determined in the heart under ischemic conditions. 

##### BAG5-HK-II Complex and BAG4

An early study in non-cardiac cells demonstrated that BAG5 interacts with and negatively regulates Parkin at the mitochondria, suppressing mitophagy [219]. A recent study in the heart further demonstrated that the association of BAG5 with mitochondria is mediated by mitochondrial HKII. Mitochondrial HKII levels are decreased in response to ischemia in the heart [220] and this results in the dissociation of BAG5 from mitochondria. The dissociation of the BAG5-HKII molecular complex leads to PINK1-independent Parkin recruitment to the mitochondria and provides protection against MI in the heart [220]. BAG5 dependent inhibition of Parkin-mediated mitophagy was also observed in other cell types including glioblastoma and neurons [220,221].

A study using a high-content genome-wide RNAi screen also identified BAG4 as a negative regulator of mitophagy induced by mitochondrial depolarization. BAG4 binds to Parkin in the cytosol and inhibits Parkin translocation to the mitochondria [222]. Thus, emerging evidence suggests that the PINK1/Parkin mediated mitophagy is regulated by BAG family proteins in a positive (BAG2, BAG3, and BAG6) or negative (BAG4 and BAG5) manner. Further studies will, however, be required to determine which BAG-dependent mechanism plays a predominant role in the regulation of mitophagy in the heart in stress conditions. 

##### Mst1

As mentioned earlier, Mst1 is an apoptotic serine/threonine kinase, contributing to the pathogenesis of ischemic heart disease. Mst1 has also been implicated in the inhibition of Parkin-mediated mitophagy. Knockdown of Mst1 in cardiomyocytes increases Parkin recruitment induced by hyperglycemia [223], and a study using Mst1 KO mice showed that Mst1 deletion reduces septic cardiomyopathy via Parkin mediated mitophagy [224]. Mst1 has also been reported to inhibit MFN-2-mediated mitophagy in cancer cells [225]. It appears that Mst1 engages diverse cellular pathways that negatively impact cell survival, including apoptosis, transcription, and mitophagy, and it will be of interest to further study how these responses regulated by Mst1 interact with each other in the heart. 

### 4.2. Ubiquitination-Independent Mitophagy

Ubiquitin-independent mitophagy mechanisms also exist, whereby BNIP3, NIX/BNIP3L, FUNDC1, BCL2L13, and cardiolipin function as LC3 receptors, targeting damaged mitochondria for autophagosomal engulfment and clearance (Figure 4). BNIP3 and its analog NIX (BNIP3L) were originally discovered as pro-apoptotic BH3-only proteins [226,227]. BNIP3 and NIX expression is induced in response to hypoxia through HIF1α activation [228,229]. NIX is also induced by Gq protein activation [22]. In addition to their apoptotic roles, BNIP3 and NIX function as mitophagy receptors [230,231,232,233], where BNIP3 and NIX have some redundancy [234]. BNIP3 overexpression in cardiomyocytes induces cell death and mitophagy, and inhibition of autophagy augments the cell death induced by BNIP3 [235]. Interestingly, BNIP3 interacts and inhibits PINK1 degradation at the mitochondria, and thereby facilitates PINK1/Parkin mediated mitophagy, suggesting crosstalk between Parkin dependent and independent mitophagy [236]. BNIP3/NIX double cardiac KO mice showed age-dependent mitochondrial abnormalities and cardiomyopathy, suggesting homeostatic roles of mitophagy regulated by BNIP3 and NIX [226]. 

The mitophagic and apoptotic effects of BNIP3 are differently regulated by the phosphorylation of BNIP3. Specifically, the phosphorylation of BNIP3 at Ser17 and Ser24 promotes its binding to LC3, facilitating mitophagy [237], while phosphorylation at Thr188 inhibits its apoptotic effect without preventing mitophagy [238]. Thus, the phosphorylation status of BNIP3 determines its pro-survival mitophagy or mitochondrial apoptosis. Similarly, it has been reported that the phosphorylation of NIX at Ser34 and Ser35 enhances autophagosome recruitment to the mitochondria [239]. The specific kinases involved in the phosphorylation of BNIP and NIX at these sites are still unknown. A recent study in myoblasts revealed that protein kinase A (PKA) phosphorylates NIX at Ser212, preventing the dimerization of NIX and inhibiting mitophagy [240,241]. The significance of phosphorylation-mediated regulation of BNIP3/NIX in cardioprotection remains to be determined. 

#### 4.2.1. FUNDC1

FUNDC1 is a MOM protein that interacts with LC3 [242,243]. Under basal conditions, FUNDC1 is phosphorylated at Tyr18 by Src [242] and at Ser13 by casein kinase 2 (CK2) [244], preventing its interaction with LC3. Phosphoglycerate mutase 5 (PGAM5), a mitochondrial phosphatase, dephosphorylates FUNDC1 at Ser13, reversing the inhibition mediated by CK2 [244]. In response to hypoxia, ULK1 is activated, translocated to the mitochondria, and phosphorylates FUNDC1 at Ser17 to stimulate mitophagy [245]. 

Accumulating evidence suggests that FUNDC1 plays an important role in cardioprotection against I/R [246,247,248]. Specifically, CK2α cardiac deletion results in the stimulation of FUNDC1-mediated mitophagy and provides protection against I/R [246]. A recent study suggests that receptor-interacting Serine/Threonine protein kinase 3 (RIPK3) inhibits FUNDC1 mediated mitophagy, enhancing cell death induced by I/R in the heart [247], although the precise mechanism by which RIPK3 inhibits FUNDC1 mediated mitophagy has not been determined. FUNDC1 expression is negatively regulated by Mst1, contributing to Mst1 mediated cell death in the heart subjected to I/R [249]. FUNDC1 is also reported to bind to inositol trisphosphate (IP_3_) type 2 receptor in mitochondria-associated endoplasmic reticulum membranes (MAMs) to maintain mitochondrial function and homeostasis in the heart [250].

#### 4.2.2. Cardiolipin

Cardiolipin is an atypical dimeric phospholipid synthesized and located in the MIM. In response to stress, cardiolipin redistributes to the MOM from the MIM and serves as an LC3 receptor to induce mitophagy in neuronal cells [251]. Cardiolipin regulates both mitophagy and apoptosis, dependent on its oxidation status: non-oxidized cardiolipin on the OMM mediates mitophagy, while oxidized cardiolipin leads to MOMP, inducing cell death [251,252]. Further studies will be required to determine which effect of cardiolipin is predominant in the heart during ischemic stress. 

#### 4.2.3. BCL2L13

Atg32 is essential for mitophagy in yeast, functioning as a mitophagy receptor. BCL2-like protein 13 (BCL2L13, also known as BCL-Rambo), a mammalian homolog of Atg32, was identified by protein database screening [253]. A recent study further demonstrated that the ULK1 complex is necessary for BCL2L13-mediated mitophagy, Upon mitophagy stimulation, BCL2L13 recruits the ULK1 complex and the interaction of LC3 and ULK1, as well as BCL2L13, is important for mitophagy [253]. BCL2L13 contains all the four BH domains and can also induce apoptotic cell death in HEK293T cells [254]. Although BCL2L13 is highly expressed in cardiomyocytes [254], the apoptotic and mitophagic effects of BCL2L13 in the heart have yet to be determined. 

### 4.3. Alternative (Non-Canonical) Autophagy

#### 4.3.1. Rab9 Dependent Mitophagy

In canonical autophagy, including mitophagy, as discussed above, ATG5, ATG7, and LC3 are required for autophagosome elongation and maturation (Figure 5A). However, a study using cells lacking ATG5 or ATG7 demonstrated that autophagic vacuoles are formed and autophagic protein degradation is induced under certain stress conditions without LC3-II formation [255], suggesting that ATG5/ATG7 independent alternative autophagy exists. ATG5/7 independent alternative autophagy requires Rab9, a small GTPase essential for membrane and protein trafficking (Figure 5A) [255]. Autophagosomes mainly originate from the endoplasmic reticulum (ER) in conventional autophagy. In contrast, alternative autophagy has been reported to originate from the *trans*-Golgi membrane. Some upstream molecules that initiate the process of autophagosome formation, such as ULK1 and Beclin1, are shared in both types of autophagy. A recent study in the heart demonstrated that alternative mitophagy is mediated by ULK1, Rab9, Rip1, and dynamin-related protein 1 (Drp1) in the heart during energy stress and is critical in the protection of the heart against ischemia [213,256]. It has also been reported that Rab9-dependent alternative mitophagy is required for IGF-II induced mitophagy in the heart. Interestingly, Parkin was shown to be involved in this process, suggesting crosstalk between canonical and alternative pathways in the heart [257]. 

#### 4.3.2. Endosome Pathway

Although the endosome is established to deliver plasma membrane proteins to lysosomes for degradation, a recent study demonstrated that the mitochondria ubiquitinated by Parkin are also eliminated by the endosome pathway in cardiomyocytes [258,259]. A ubiquitinated mitochondrion is captured by ESCRT complexes on the Rab5 positive early endosome and, after its maturation, the endosome fuses with the lysosome for degradation (Figure 5B). The endosomal pathway is activated prior to the onset of conventional mitophagy, suggesting that this pathway is temporally distinct and is responsible for the initial phase of mitochondrial clearance during stress in cardiomyocytes. Additionally, the endosomal pathway is also utilized by the mitophagy receptor BNIP3, independent of Parkin [259]. These findings indicate that redundancy exists in the degradation pathways to ensure efficient mitochondrial clearance. 

#### 4.3.3. Mitochondrial-Derived Vesicles (MDVs)

Mitochondria labeled by the PINK1/Parkin pathway are also subjected to mitochondrial-derived vesicles (MDVs)-dependent elimination [260,261]. MDVs are membranous structures containing damaged cargo that bud off and are secreted and shuttled to lysosomes via the endosomal system independently of the autophagic protein ATG5 or LC3 [262] (Figure 5C). This pathway is independent of mitochondrial depolarization but is induced by mitochondrial ROS. MDVs are continuously generated under basal conditions and are also induced rapidly in response to oxidative stress. Thus, the MDV pathway has been demonstrated to play a homeostatic role as well as to function as a first response to stress in cardiomyocytes [261,263,264]. Further studies will be required to determine the functional relevance of the endosomal and MDV pathways in the in vivo heart. 

## 5. Mitochondrial Fission and Fusion

Mitochondria are highly dynamic organelles that constantly undergo fusion and fission in response to environmental cues. Mitochondrial fusion is induced by mitofusins (Mfn1 and Mfn2) and optic atrophy 1 (OPA1), resulting in tubular and elongated mitochondria, while fission (division) is regulated by Drp1, Fis1, MFF, and Mid49/51 resulting in small mitochondria [154,265,266,267,268]. Mitochondrial fusion and fission are critical in maintaining basal cardiac homeostasis, as evidenced by the observations that cardiac-specific deletion of Drp1 or Mfn1/2 causes cardiac dysfunction [157,269,270]. Mitochondrial fission is regulated by Drp1, a cytosolic GTPase that can translocate to the mitochondria to generate the force necessary for mitochondrial fission. Fis1, MFF, and Mid19/51 function as receptors for Drp1 at the mitochondria. Mitochondrial translocation of Drp1 can be increased or decreased through post-translational modification, most clearly phosphorylation but also ubiquitination, SUMOylation, and S-nitrosylation [271,272]. Two phosphorylation sites on Drp1, Ser616, and Ser637, are well characterized. Mitochondrial distribution of Drp1 and mitochondrial fission are induced by phosphorylation of Drp1 at Ser616 but are inhibited by phosphorylation of Drp1 at Ser637. In cardiomyocytes, phosphorylation of Ser616 has been shown to be mediated by cyclin-dependent kinase 1, PKCδ, CaMKII, ERK, RIP1, and ROCK [213,273,274,275,276]. Phosphorylation of Drp1 at Ser637, an inhibitory site, has been shown to be regulated by protein kinase A (PKA), PKD, and Pim1 in cardiomyocytes [277,278,279].

Mitochondrial dynamics are functionally related to the mitochondrial quality control mechanisms and thus also play an important role in adaptation to stress conditions. While symmetrical fission results in two equal daughter mitochondria to replicate and expand the cellular mitochondrial pool, asymmetrical fission is induced in response to mitochondrial stress and facilitates the segregation of damaged organelle components which are in turn sequestrated and degraded by mitophagy [266,267,272,280]. However, the effects of the global stimulation of mitochondrial fusion or fission on cellular survival appear to be context-dependent and somewhat controversial [157,269,272,281,282,283,284,285,286]. For instance, pharmacological inhibition of Drp1 has been shown to be protective against I/R in murine hearts [282,286,287], and a study using heterozygous Drp1 knockout mice (Drp1^+/-^) showed that I/R injury is decreased in Drp1^+/-^ mouse hearts [288], suggesting that Drp1-mediated mitochondrial fission contributes to I/R injury. On the contrary, cardiac-specific Drp1 heterozygous KO mice showed increased I/R injury with an accumulation of ubiquitinated dysfunctional mitochondria, suggesting the protective role of mitochondrial fission through the removal of damaged mitochondria by mitophagy [269]. As mentioned earlier, Drp1 mediated fission also contributes to the cardioprotection mediated by Rab9 dependent alternative mitophagy [213,256]. The protective role of mitochondrial fission is further supported by a study on the cardiac-specific ablation of both Mfn1 and Mfn2. In cardiac-specific double KO mice, mitochondria are smaller and the infarct size after in vivo I/R is reduced compared with WT mice [281]. Although the reason for these contradictory results is not clear, these studies suggest the close interaction between mitophagy and mitochondrial fusion/fission but further studies will be required to determine exactly how mitochondrial dynamics coordinate with the molecular machinery of mitophagy to segregate the damaged portion of mitochondria for removal. A recent study using focused ion beam scanning electron microscopy revealed that multiple sub-networks of mitochondria exist in adult ventricular myocytes and that malfunctioning mitochondria undergo separation from the network to minimize the spread of dysfunction throughout the entire network [289]. This supports the concept that the segregation of damaged mitochondria is an asymmetric and spatially localized event induced in response to stress in the heart. Thus, it would be particularly important to understand how symmetric vs. asymmetric mitochondrial fission are differentially regulated under basal and stress conditions. 

## 6. Conclusions and Future Perspectives

Mitochondria are the essential organelle producing cellular energy and regulating cell fate, and mitochondrial dysfunction is commonly observed in the failing heart. Thus, the protection and quality control of mitochondria under stress conditions are critical in maintaining cardiac homeostasis to prevent the development of heart failure. Our understanding of the molecular machineries that regulate mitochondrial cell death and mitophagy has been advanced significantly, and accumulating evidence suggests that these cellular processes are tightly regulated by intracellular signaling molecules. Developing drugs/agents that are capable of preventing cardiomyocyte loss in the heart is of significant interest to create pharmacological interventions that preserve the integrity of mitochondria through the modulation of these signaling molecules may provide a means of treating heart failure. However, much of our knowledge to date comes from studies performed in vitro. Indeed, despite the extensive experimental evidence, the transition of mitochondria protective interventions into the clinical setting for ischemic heart disease has been challenging. Cs-A, a desensitizer of the mPTP, has been shown to be protective in an initial small proof-of-concept clinical study in acute myocardial infarction [290]. However, subsequent large clinical trials failed to demonstrate the beneficial effects of Cs-A [291,292]. Similarly, although it is generally accepted that ROS scavenging molecules exert protective effects, attempts to limit ROS production with generic antioxidants have proven to be ineffective in large clinical trials [293]. The lack of protective effect could be explained by insufficient mitochondrial targeting within the ischemic myocardium. Emerging evidence suggests that the cardioprotective effects of interventions could be improved using a drug delivery system (DDS) including nanoparticle and mitochondria-targeting moieties [294,295,296]. Studies in rodents have suggested that the mitochondria-targeting delivery of Cs-A and antioxidants have improved cardioprotective efficacy [295,296]. Further studies in large animals and humans are, however, required to elucidate the relevance of the mitochondria protective signaling pathways in ischemic heart disease. 

## Figures and Tables

**Figure 1 cells-10-03330-f001:**
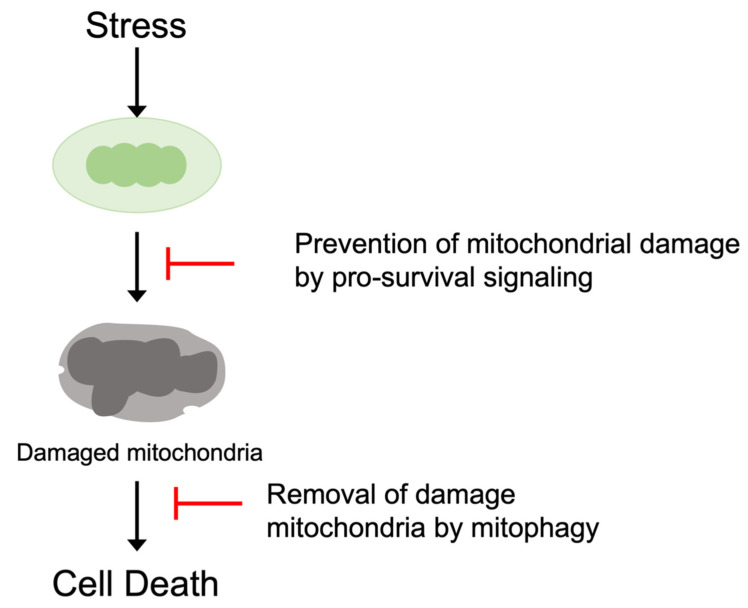
Prevention of the mitochondrial death pathway. Pro-survival signaling pathways antagonize the induction of the mitochondrial death pathway. Damaged mitochondria are also eliminated by mitophagy to preserve mitochondrial integrity and inhibit cell death.

**Figure 2 cells-10-03330-f002:**
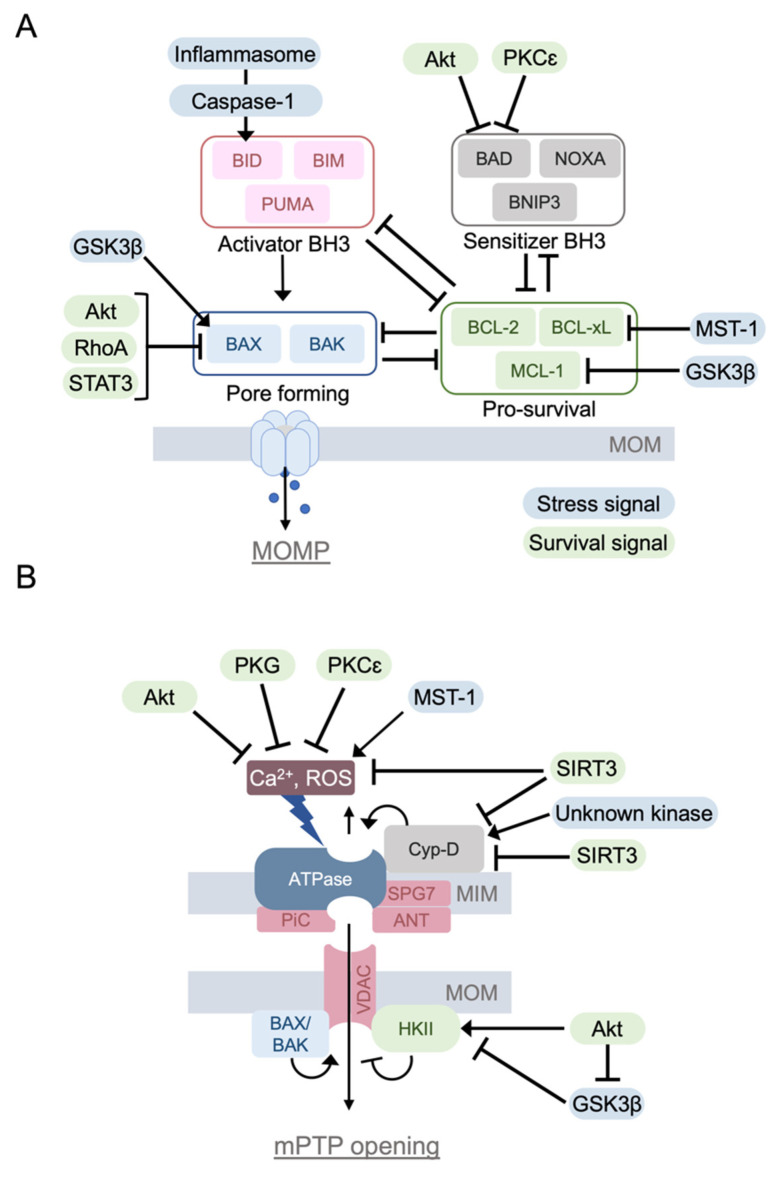
The BCL-2 family proteins and the mitochondrial permeability transition pore (mPTP). (**A**) BCL-2 family proteins regulate mitochondrial outer membrane permeabilization to induce apoptotic cell death, which is positively and negatively regulated by intracellular signaling molecules. (**B**) mPTP is a large conductance channel that is activated by Ca^2+^ overloading and ROS, inducing the rupture of mitochondrial membranes and eventual cell death. Although the molecular constituents of the mPTP continue to be disputed, it has been shown that the opening of the mPTP is regulated by intracellular signaling molecules.

**Figure 3 cells-10-03330-f003:**
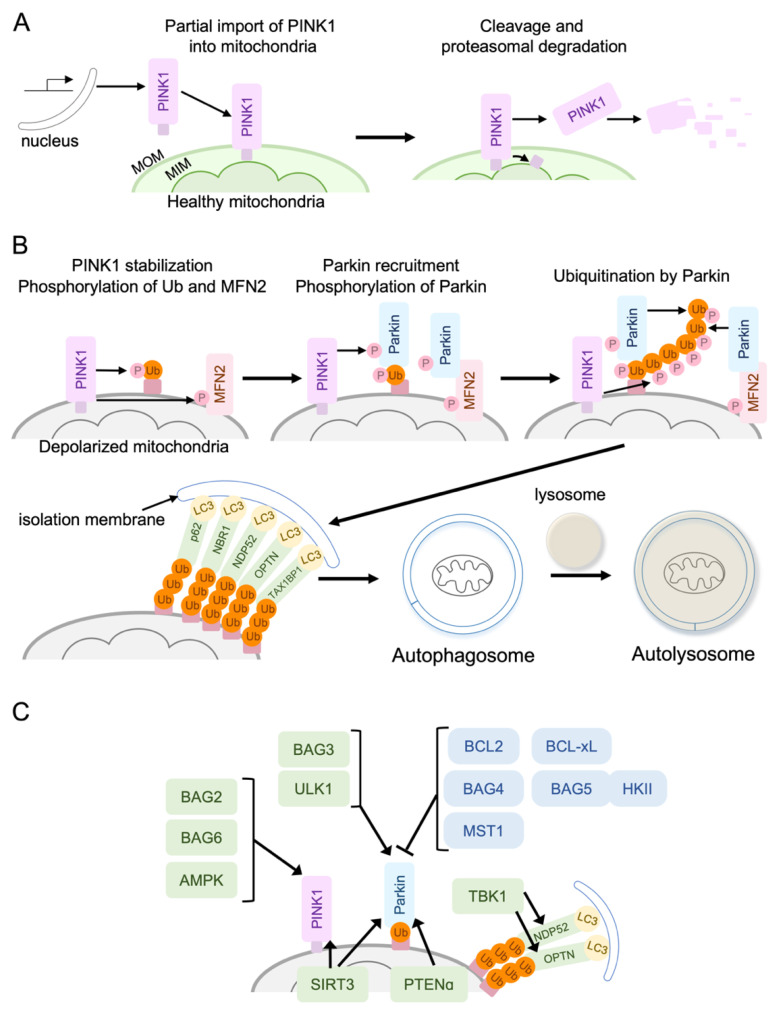
The PINK1/Parkin-mediated mitophagy. (**A**) In healthy cells, PINK1, a mitochondrial serine/threonine kinase, is cleaved by an inner-mitochondrial membrane protease, and the cleaved PINK1 is then retro-translocated to the cytosol where it undergoes proteasomal degradation. (**B**) Under stress conditions, dissipation of mitochondrial membrane potential leads to the inhibition of PINK1 cleavage and supports the accumulation of full-length PINK1 at the MOM. PINK1 phosphorylates basal MOM ubiquitin which drives Parkin recruitment to the mitochondria and further phosphorylates Parkin to stabilize the binding of Parkin to ubiquitin. In addition, PINK1 phosphorylates mitofusin-2 (MFN2), a mitochondria fusion protein, which also serves as a receptor for Parkin at MOM. Ubiquitinated mitochondria are recognized by the autophagosome membrane through the autophagy receptor proteins, which interact with LC3 on the autophagosome membrane. (**C**) Signaling molecules regulating the PINK1/Parkin pathway.

**Figure 4 cells-10-03330-f004:**
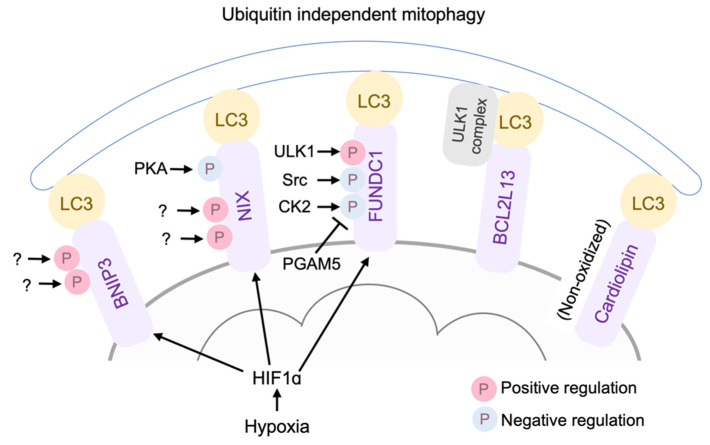
Ubiquitin independent mitophagy mechanisms. BNIP3, NIX/BNIP3L, FUNDC1, BCL2L13, and cardiolipin function as LC3 receptors, targeting damaged mitochondria for autophagosomal engulfment and clearance.

**Figure 5 cells-10-03330-f005:**
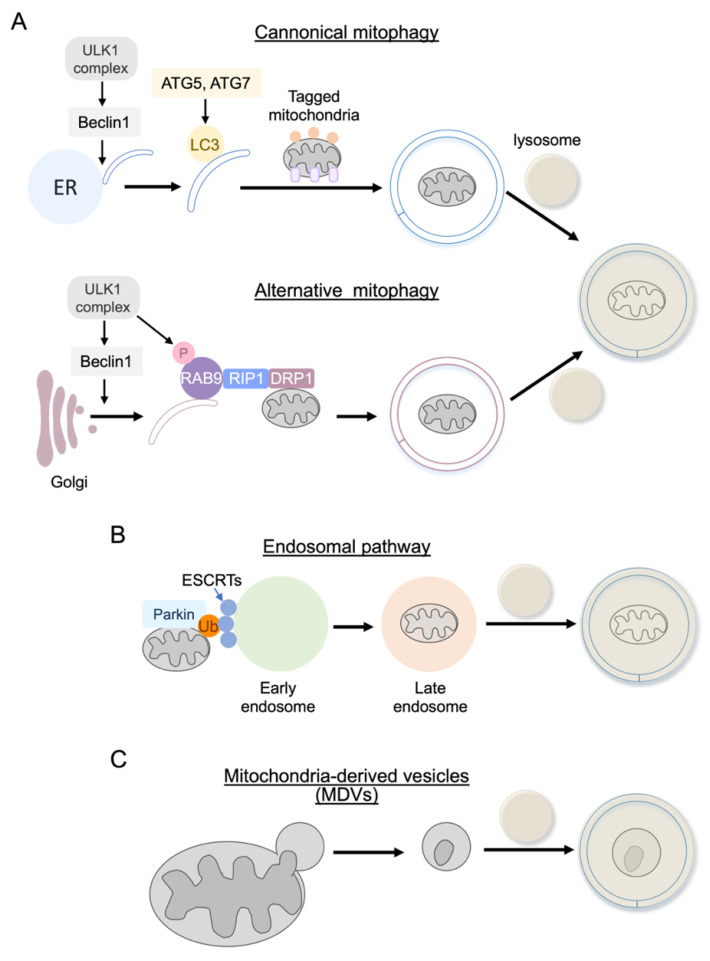
Alternative (non-canonical) autophagy. (**A**) Rab9-dependent mitophagy is not dependent on ATG5/ATG7 and LC3, which are required for canonical autophagy, including PINK1/Parkin-mediated mitophagy. Rab9-dependent autophagy is mediated by ULK1, Rab9, Rip1, and dynamin-related protein 1 (Drp1) in the heart. (**B**) Endosomal pathway. Mitochondria ubiquitinated by Parkin are also eliminated by the endosome pathway. A ubiquitinated mitochondrion is captured by ESCRT complexes on the Rab5 positive early endosome and, after its maturation, the endosome fuses with the lysosome for degradation. (**C**) Mitochondrial-derived vesicles (MDVs). Mitochondria labeled by the PINK1/Parkin pathway are also subjected to MDVs-dependent elimination.

## Data Availability

Not applicable.

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
