# Peer review of "Molecular Signaling to Preserve Mitochondrial Integrity against Ischemic Stress in the Heart: Rescue or Remove Mitochondria in Danger"

_cells, 2021, doi:10.3390/cells10123330_

Round 1

Reviewer 1 Report

Dear Editor,
The review paper presented by Yu and Miyamoto is a detailed and well-written review on the mechanisms of regulation of mitochondrial functions in cardiac physiology and pathology.
The authors have compiled a range of information on various underlying pathways of mitochondrial function and structure.
Although the review contains valuable information, it remains, in a general view, a text about mitochondrial biology and can not yet link the explained findings to the objective of review namely: ischaemic stress.
I am sure that the manuscript fulfils the authors' vision and Cells readers´ expectations when the authors discuss specific proteins or given signalling pathways in the context of the stressed heart...this can be easily achieved by adding at the end of each main section some examples of available studies about ischemic stress in the heart.

I assume that there are studies dealing with mitochondria as a therapeutic target for such cardiac pathology, it would be great if authors can add this information to the text as well.

Yours sincerely

Author Response

The review paper presented by Yu and Miyamoto is a detailed and well-written review on the mechanisms of regulation of mitochondrial functions in cardiac physiology and pathology.

The authors have compiled a range of information on various underlying pathways of mitochondrial function and structure.

  1. Although the review contains valuable information, it remains, in a general view, a text about mitochondrial biology and cannot yet link the explained findings to the objective of review namely: ischaemic stress.

I am sure that the manuscript fulfils the authors' vision and Cells readers´ expectations when the authors discuss specific proteins or given signaling pathways in the context of the stressed heart...this can be easily achieved by adding at the end of each main section some examples of available studies about ischemic stress in the heart.

Response: Thank you for the suggestion. Now we discuss previous studies that have demonstrated the significance of the molecules in the context of cardiac damage induced by ischemic stress. As suggested, we added these descriptions at the end of each section.

Please see

Lines 103-105,

Lines 110-112,

Lines 138-142,

Lines 168-173,

Lines 212-215,

Lines 239-245,

Lines 265-269,

Lines 289-296,

Lines 308-311,

Lines 324-328,

Lines 363-367,

Lines 377-381,

Lines 397-428,

Lines 482-486,

Lines 492-493,

Lines 505-508,

Lines 521-525,

Lines 564-571,

Lines 677-678,

Lines 766-768,

Lines 802-833.

  1. I assume that there are studies dealing with mitochondria as a therapeutic target for such cardiac pathology, it would be great if authors can add this information to the text as well.

Response: We really appreciate your suggestion. This is an excellent point. Unfortunately, the transition of mitochondria protective interventions into the clinical setting for ischemic heart disease has been challenging. Thus we have discussed this issue at the end of our manuscript in the "Conclusion and future perspectives" (lines 885-901).

We are grateful for the constructive critiques from the reviewer.

Reviewer 2 Report

The submitted article is well written, the structure of the chapters and headings is logical, provides a detailed review on molecular signalling of mitochondrial quality control an dynamics. In accordance with the title we find that the role of the aforementioned processes regarding the ischemic stress in the heart has been addressed properly although we felt it should be brought out more clearly how the fission-fusion mechanisms (last chapter seems a bit short and less detailed in comparison to the previous headings) are involved by cite few recent in vivo studies on modulation of mitochondrial quality control processes.

Author Response

The submitted article is well written, the structure of the chapters and headings is logical, provides a detailed review on molecular signaling of mitochondrial quality control and dynamics. In accordance with the title we find that the role of the aforementioned processes regarding the ischemic stress in the heart has been addressed properly although we felt it should be brought out more clearly how the fission-fusion mechanisms (last chapter seems a bit short and less detailed in comparison to the previous headings) are involved by cite few recent in vivo studies on modulation of mitochondrial quality control processes.

Response: We appreciate your suggestion. We have added more details regarding the relationship between mitochondrial dynamics and mitophagy. As suggested, we discuss recent in vivo studies (lines 795-826). Unfortunately, it is still not entirely clear whether mitochondrial fission (or fusion) enhances mitophagy or elicits deleterious effects in the heart subjected to ischemic stress.

We are grateful for the constructive critiques from the reviewer.